# Cross-Cultural Adaptation, Reliability, and Psychophysical Validation of the Pain and Sleep Questionnaire Three-Item Index in Finnish

**DOI:** 10.3390/jcm10214887

**Published:** 2021-10-23

**Authors:** Jani Mikkonen, Ville Leinonen, Hannu Luomajoki, Diego Kaski, Saana Kupari, Mika Tarvainen, Tuomas Selander, Olavi Airaksinen

**Affiliations:** 1Private Practice, Helsinki, Finland; 2Department of Surgery (Incl. Physiatry), Institute of Clinical Medicine, University of Eastern Finland, 70211 Kuopio, Finland; Olavi.Airaksinen@kuh.fi; 3Institute of Clinical Medicine-Neurosurgery, University of Eastern Finland, 70211 Kuopio, Finland; ville.leinonen@kuh.fi; 4Department of Neurosurgery, Kuopio University Hospital,70211 Kuopio, Finland; 5ZHAW School of Health Professions, Zurich University of Applied Sciences, CH-8401 Winterthur, Switzerland; luom@zhaw.ch; 6Department of Clinical and Movement Neurosciences, University College London, London WC1E 6BT, UK; d.kaski@ucl.ac.uk; 7Department of Applied Physics, University of Eastern Finland, 70211 Kuopio, Finland; saanaku@uef.fi (S.K.); mika.tarvainen@uef.fi (M.T.); 8Department of Clinical Physiology and Nuclear Medicine, Kuopio University Hospital, 70211 Kuopio, Finland; 9Science Service Center, Kuopio University Hospital, 70211 Kuopio, Finland; tuomas.selander@kuh.fi; 10Department of Physical and Rehabilitation Medicine, Kuopio University Hospital, 70211 Kuopio, Finland

**Keywords:** pain, chronic low back pain, sleep, questionnaire, cross-cultural validation, patient-reported outcome measure, postural control, dizziness, actigraphy, sleep quality

## Abstract

Reciprocal relationships between chronic musculoskeletal pain and various sleep disturbances are well established. The Pain and Sleep Questionnaire three-item index (PSQ-3) is a concise, valid, and reliable patient-reported outcome measure (PROM) that directly evaluates how sleep is affected by chronic low back pain (CLBP). Translation and cross-cultural validation of The Pain and Sleep Questionnaire three-item index Finnish version (PSQ-3-FI) were conducted according to established guidelines. The validation sample was 229 subjects, including 42 pain-free controls and 187 subjects with chronic musculoskeletal pain. Our aims were to evaluate internal consistency, test–retest reliability, measurement error, structural validity, convergent validity, and discriminative validity and, furthermore, to study the relationships between dizziness, postural control on a force plate, and objective sleep quality metrics and total PSQ-3-FI score. The PSQ-3-FI demonstrated good internal consistency, excellent test–retest reliability, and small measurement error. Confirmatory factor analysis confirmed acceptable fit indices to a one-factor model. Convergent validity indicated fair to good correlation with pain history and well-established pain-related PROMs. The PSQ-3-FI total score successfully distinguished between the groups with no pain, single-site pain, and multisite pain. A higher prevalence of dizziness, more impaired postural control, and a general trend towards poorer sleep quality were observed among subjects with higher PSQ-3-FI scores. Postural control instability was more evident in eyes-open tests. The Finnish PSQ-3 translation was successfully cross-culturally adapted and validated. The PSQ-3-FI appears to be a valid and reliable PROM for the Finnish-speaking CLBP population. More widespread implementation of PSQ-3 would lead to better understanding of the direct effects of pain on sleep.

## 1. Introduction

Chronic low back pain (CLBP) is the leading disability globally [1]. More than half of patients with CLBP experience various sleep disturbances, such as problems falling asleep and staying asleep, waking up because of pain, difficulties getting back to sleep after awakening, restless sleep, fatigue after sleeping, insomnia, and/or restless legs syndrome [2,3]. Sleep disturbances have a fundamental effect on health and are associated with mental disorders such as anxiety; depression; numerous chronic systemic metabolic, cardiovascular, respiratory, and neurological diseases; and increased risk of certain types of cancer [4,5,6], as well as negative effects on short-term, day-to-day function, and well-being [7,8]. Reciprocal relationships between sleep disturbances and different chronic musculoskeletal pain conditions have also been well-established [9,10,11]; thus, there is a need for a concise, reliable, and valid patient-reported outcome measure (PROM) for clinical assessments and research to measure the direct effects of pain on sleep.

The Pain and Sleep Questionnaire three-item index (PSQ-3), a three-question questionnaire, was developed in 2012 [12] to directly assess the impact of pain on sleep during a one-week period. The three items are “1. How often have you had trouble falling asleep because of pain?”, “2. How often have you been awakened by pain during the night?”, and “3. How often have you been awakened by pain in the morning?”. The possible answers range on a scale from 0 indicating “never” to 100 representing “always”. Previous validation of the PSQ-3 demonstrated good internal consistency and good structural validity for a one-factor model, but doubtful convergence validity [12,13].

A 2018 systematic review of PROMs for clinical assessment of sleep quality among patients with chronic pain reviewed twelve different questionnaires assessing sleep on six different pain populations [13]. The PSQ-3 has been validated for the CLBP population [12] and the Chronic Pain Sleep Inventory for patients with hip osteoarthritis [14]. Interestingly, these PROMs appear essentially identical, as the Chronic Pain Sleep Inventory includes three questions that are the same as the questions in the PSQ-3. There have been no previous cross-cultural validations of the PSQ-3 or Chronic Pain Sleep Inventory. Overall, only one cross-cultural validation has previously been performed for the twelve PROMs for the effect of pain on sleep [13].

Sleep quality and disturbances have several significant effects on daytime functions, such as increased postural control instability [15,16,17] and subjective symptoms such as dizziness [18]. Postural control instability can be objectively studied using a force plate, which is a mechanical sensing system designed to measure the ground reaction forces and moments involved in postural control [19]. Diagnosis of dizziness is mostly based on patient-reported symptoms, and there is no single objective clinical test to diagnose or classify dizziness into different subtypes [20,21]. Due to previous studies showing associations between sleep disturbances and postural control instability and dizziness, we hypothesized that there may be potential relationships between higher PSQ-3 scores and postural instability and an increased prevalence of subjective dizziness. Despite the known effects of poor sleep on balance and vestibular function, the relationships between PSQ-3 scores and postural stability and subjective dizziness have never been formally explored.

Objective sleep quality can be directly assessed via actigraphy, which monitors activity and rest cycles based on movement (accelerometer) data [22]. Sleep-wake patterns are evaluated by determining activity counts using scoring algorithms [22,23]. Frequently used measures of the continuity of sleep in various conditions include the total sleep time, sleep efficiency, and amount and duration of awakenings [24]. Previous studies have reported associations between various other pain–sleep questionnaires and actigraphy measurements among subjects with chronic musculoskeletal pain [2,25,26]; however, the relationship between actigraphy measurements and the PSQ-3 has not yet been studied. Hence, actigraphy measurements were also included in this study.

The study aimed to translate and cross-culturally adapt the PSQ-3 into Finnish (PSQ-3-FI) and to evaluate its reliability (internal consistency, test–retest reliability, and measurement error), structural validity for a one-factor model, convergence validity (based on its correlation with pain history, prevalence of dizziness history, The Central Sensitization Inventory (CSI), The Tampa Scale of Kinesiophobia (TSK), The Depression Scale (DEPS), the 5-level EQ-5D version of the EuroQol (EQ-5D-5L), and The Roland–Morris Disability Questionnaire (RMDQ)), and discriminative validity. We also investigated the relationships between the PSQ-3-FI and subjective dizziness, postural control on a force plate and objective sleep quality.

## 2. Materials and Methods

Ethical approval for this study was obtained from the Research Ethics Committee of the Northern Savo Hospital District (identification number, 1106/13.02.00/2018). Written informed consent was obtained from all subjects before the study began, and the study was conducted in accordance with the Declaration of Helsinki. This validation study adhered, where applicable, to the Consensus-based Standards for the selection of health Measurement Instruments (COSMIN) checklist to ensure the methodological quality of studies on measurement instruments [27,28].

### 2.1. Study Subjects

The subjects for this study were recruited via advertisements on the website of the private chiropractic practice where this study was performed, as well as the websites and social media accounts of a variety of national Finnish musculoskeletal pain and spine-related organizations and colleagues involved in healthcare. All subjects from the general population who met the study inclusion criteria were invited to participate, regardless of whether they experienced pain or not. The inclusion criteria were: (1) age 18 to 65 years old and (2) proficient in written and spoken Finnish. The exclusion criteria were: (1) a history of cancer or (2) a history of trauma or conditions involving the central nervous system, including dementia, Alzheimer’s disease, and multiple sclerosis. A total of 257 subjects provided informed consent to participate in the study and booked a clinical appointment using the online booking system.

### 2.2. Translation and Cross-Cultural Adaptation of the PSQ-3

Translation and cross-cultural validation of the Finnish CSI were conducted following standard guidelines and included forward–backward translation [29]. Permission to translate the PSQ-3 into Finnish was granted by the first author of the study, who identified the one-factor structure of the PSQ-3 [12]. The PSQ-3 was initially translated from English into Finnish by the first author (JM; completed undergraduate and postgraduate degrees in English-speaking countries) and a professional translator specializing in medical and healthcare texts, both of whom are native Finnish speakers and were blinded to the other’s translations. Then, an expert panel composed of the second (VL) and third (HL) authors independently reviewed the initial translations, selected the most appropriate translations, and suggested and discussed changes for one item with the first author. A small number of minor changes in wording were made. Next, the translated version was back-translated by another native English-speaking professional translator who is fluent in Finnish; this translator was naïve to the purpose of this study and the PSQ-3. The backtranslation was assessed and approved by the author of the original English version of the PSQ-3, who is a native English speaker, and some final changes in the wording of the content were made.

Finally, the face validity of the provisional PSQ-3-FI was assessed among twenty subjects with chronic musculoskeletal pain, who were informed of the purpose of this study and were requested to provide non-structured verbal or written feedback on each item on the provisional PSQ-3-FI. All subjects provided positive feedback on their comprehension of each of the items of the PSQ-3-FI and completed the questionnaire without difficulties. The Finnish version of PSQ-3 can be found in Appendix A.

### 2.3. Data Collection

Data were collected from May 2019 until March 2020 at a single private chiropractic practice. During each individual’s clinical visit, objective clinical measurements of postural control were obtained using a force plate. After returning home, subjects were instructed to complete an online web-based form to collect demographic information (age, gender, weight, height, and pain history) and to fill in clinical questionnaires, including the PSQ-3-FI. During the data analysis phase, body mass index was calculated for each subject based on their self-reported height and weight. The total sample size for PSQ-3-FI validation was determined by the ratio of the number of items in each measure, which was 76.7 and hence exceeded recommended range from 2 to 20 items in each measure [30]. To assess test–retest reliability, the subjects were emailed and invited to complete the PSQ-3-FI again 7 ± 1 days after completing the initial questionnaires. The email invitations were stopped after 104 subjects had completed the PSQ-3-FI twice. The ratio of the number of items in each measure to the sample size for evaluating test–retest reliability was 1:34.7. Therefore, the recommended 1:5 ratio was satisfied [31]. All test–retest participants were asked to avoid starting any new types of pain medication and/or physical treatment, when ethically possible, during the 7 ± 1-day gap between administration of the tests. Actigraphy data were collected between December 2019 to March 2020 and August 2020 to November 2020; 24 h actigraphy data were always collected from a Tuesday afternoon until the next Wednesday. The break in actigraphy data collection was due to the COVID-19 outbreak in Finland.

#### 2.3.1. Subject-Reported Pain History Questions

Each subject completed a structured web-based pain history, which asked dichotomous (yes/no) questions related to the presence of chronic low back pain, referred pain to leg or leg pain (if “yes” to chronic low back), presence of other chronic musculoskeletal pain, chronic headaches, and history of a rheumatic disease previously diagnosed by a physician. Subjects who reported pain were also asked to rate the severity of their pain on a numerical pain scale ranging from 0–10 and to indicate the duration of pain in months.

#### 2.3.2. Patient-Reported Clinical Outcome Measures

The three-item PSQ-3 sleep questionnaire was designed to measure the impact of chronic pain on sleep over the previous seven days [12]. The three questions are “1. How often have you had trouble falling asleep because of pain?”, “2. How often have you been awakened by pain during the night?”, and “3. How often have you been awakened by pain in the morning?”. We translated these items into Finnish for this study. The original PSQ-3 employed a visual analog scale that ranges from 0 to 100 mm. However, due to the difficulty of representing a visual analog scale in a universal digital format, we adopted a numerical eleven-point rating scale (NRPS) from 0 to 10 for the Finnish version. In both the original and Finnish versions, 0 indicates “never” and 100 mm or 10 on the numerical scale represents “always.” Thus, the final score for the PSQ-3-FI ranges from 0 to 30 rather than 0 to 300.

The eleven-point numerical pain scale (NPRS) assesses pain on a scale ranging from 0 (no pain at all) to 10 (worst pain imaginable) [32]. Chronic pain was defined as more than three days of pain every week for more than three months.

The Central Sensitization Inventory (CSI) questionnaire contains two parts [33]. Part A is composed of 25 questions in which the frequency of CS-related symptomology is rated on a Likert-like scale from 0 (never) to 4 (always). The total score ranges from 0 to 100; higher scores indicate a higher frequency and number of CS-related symptoms. [34]. Part B includes “No/Yes” and “year diagnosed” questions about previous diagnoses of ten central sensitization syndromes or related diagnoses; Part B is not scored [35]. The CSI was previously translated into Finnish and validated in the Finnish population [36].

The 17-item Tampa Scale of Kinesiophobia (TSK) is used to assess subjective kinesiophobia (fear of movement). Each item is rated as: 1 = strongly disagree, 2 = disagree, 3 = agree, or 4 = strongly agree. The possible scores range from 17 to 68; higher scores indicate more severe kinesiophobia [37]. The TSK was previously translated into Finnish and validated in the Finnish population [38].

The 10-item Depression Scale (DEPS) was designed to assess depressive symptoms. Each item response is rated on a four-point Likert scale as: 0 = not at all, 1 = a little, 2 = quite a lot, or 3 = extremely. Higher scores (range, 0 and 30) indicate a higher possibility of a diagnosis of a major depressive disorder [39]. The DEPS has been validated for patients with CLBP [40].

Health-related quality of life was assessed using the Finnish translation of the 5-level EQ-5D version of the EuroQol (EQ-5D-5L) questionnaire [41], which provides a simple descriptive profile of a respondent’s health status over five dimensions: mobility, self-care, usual activities, pain/discomfort, and anxiety/depression. Each dimension is rated as: 0 = no problems, 1 = slight problems, 2 = moderate problems, 3 = severe problems, or 4 = unable to/extreme problems. The EQ-5D-5L also includes the EQ visual analog scale (EQ VAS), which assesses the respondent’s current overall health status using a visual, vertical analog scale that ranges from 0 (dead) to 1 (full health) [41]. The index value between 0 and 1 is calculated. A standard value set has not yet been defined for the Finnish population; therefore, as recommended by the EuroQol EQ-5D-5L User Guide, the index values for this study were calculated using a Danish value set [42].

The 24-item Roland–Morris Disability Questionnaire (RMDQ) is an extensively validated questionnaire of disability among patients with chronic low back pain [43]. The RMDQ score is obtained by summing up the number of low-back-pain-related daily activity disabilities to which the respondents check “yes”. A higher total score (range, 0 to 24) suggests a higher extent of low back pain related-disability [44].

#### 2.3.3. Subjective Dizziness Structured Interview

In agreement with the literature, where dizziness is based on symptoms rather than a clinical diagnosis of a specific vestibular or neuromusculoskeletal condition [45], structured interviews were conducted during the clinical visit to assess the subject’s history of dizziness in the previous year. The questions were “Do you suffer dizziness at the moment? Have you suffered dizziness during the last 12 months? Dizziness means an abnormal sensation causing disability for more than one day, which is not the same as normal brief light-headedness when standing up quickly”. We asked further questions to all subjects who had experienced dizziness resulting in disability that had persisted for more than 24 h (*n* = 52; 23%) to classify the symptoms of dizziness into seven categories: 1. off balance or unsteadiness, 2. Light-headedness, 3. feeling as if passing out, 4. spinning or vertigo, 5. floating or tilting sensation, 6. blurring of vision when moving the head, or 7. other types. This classification was based on recent dizziness subtype research [45].

#### 2.3.4. Clinical Tests of Postural Control Using a Force Plate

The cohort of subjects who reported pain was divided into two groups according to their PSQ-3-FI score. As there are no established PSQ-3 cut-off scores for different severity classes of the effect of pain on sleep, we classified the subjects on two groups based on a PSQ-3-FI score of 4 or less (cumulative 48%; *n* = 110) or a score of 5 or more (cumulative 52%; *n* = 119).

Postural stability was measured with a four-channel portable computerized force plate (BT4; HUR Labs Oy, Tampere, Finland). As the subjects completed the questionnaires after their clinical visit, the assessor was blinded to the participants’ pain history and questionnaire scores. Postural control measurements included length, area, and velocity of center of pressure (COP) displacement, which are the most commonly used parameters for postural control in previous quality of sleep and postural control-related studies [15,16]. Various postural control parameters describe the neuromuscular response to shifts in the body’s center of mass measured on the force plate [19].

The postural control tests were carried out in the same room under identical conditions for all subjects, including the distance to the opposite wall and lighting. The force plate was calibrated before each individual’s measurement. All subjects were instructed to stand barefoot, with their feet as close together as comfortably possible. If the subjects found this stance unnatural, they were instructed to place their feet farther apart to create a more stable and natural-feeling standing stance. Small variations in foot stance should not affect the results of bipedal balance tests [46]. The subjects were instructed to look straight ahead and try to maintain a steady posture in a relaxed manner, with their arms at their sides in a relaxed position. There was no clear fixation point for gaze, and the opposite wall was more than three meters away.

Four postural control tests were carried out in the same non-randomized order: eyes open on a stable surface (EOS), eyes closed on a stable surface (ECS), eyes open on an unstable foam surface (EOU), and eyes closed on an unstable foam surface (ECU). The protocol for the bipedal standing tests was similar to the Modified Clinical Test of Sensory Interaction in balance (CTSIB-M) protocol, except each test lasted 60 s and was conducted once. Sixty seconds is the most commonly used measurement time for the bipedal standing test for the CLBP population [47]. The CTSIB-M has been shown to be a reliable, valid test in adults with vestibular disorders [48]. No similar protocol has been recommended or validated for the CLBP population [49]. A five-second pre-phase period was employed before the actual COP measurement of 60 s. Additionally, there was a short designated resting period between each test, when the instructions for the next test were repeated, and the subjects had to step off the postural plate between the second and third tests to allow the balance pad to be placed on the plate. The sampling frequency was set to 50 Hz, as recommended by the manufacturer, to obtain a balance between consistent data acquisition and manageable data size. A rectangular, high-density (50 kg/m^3^) closed-cell Airex Balance Pad (delivered by the manufacturer with the force plate) was used in all tests requiring a foam surface to provide an unstable surface.

#### 2.3.5. Sleep Quality Recordings

Sleep activity was measured with a ActiGraph GT9X link research-grade activity bracelet (ActiGraph LLC., Pensacola, FL, USA) over 24 h and the data were analyzed with Actilife 6.0 analysis software (ActiGraph LLC., USA). The following five parameters were selected to represent sleep quality: 1. total sleep time, 2. sleep efficiency, i.e., the ratio between the total sleep time and time spent in bed, 3. the number of awakenings lasting more than one minute, 4. average awakening length, and 5. the number of awakenings greater than or equal to five minutes.

PSQ-3 validation data were collected simultaneously with data for validation of the Finnish version of the CSI. The subjects with the lowest and highest CSI and PSQ-3-FI scores were also invited to participate in this study. The recruitment process was stopped when the required 40 subjects were recruited and both groups included an almost equal number of subjects. The recruited subjects were divided into two groups based on their PSQ-3 scores: (1) a group with PSQ-3 scores ≤4 (*n* = 19) and (2) a group with scores ≥5 (*n* = 21). Similar numbers of subjects were assessed in previous studies of comparable subject cohorts to compare activity measures between two groups [50,51].

### 2.4. Statistical Methods

Statistical analysis was performed using SPSS version 25 (IBM SPSS Statistics for Windows, Version 25.0. IBM Corp., Armonk, NY, USA), R statistical software version 4.0.4 was used for factor analysis, and sleep quality analysis was conducted using MATLAB (R2019b, MathWorks, Natick, MA, USA). Statistical significance was defined as *p* < 0.05. Data are reported as percentages or means with standard deviations (mean ± SD). Cronbach’s alpha was used to assess internal consistency; an alpha value between 0.70 and 0.90 was considered good, and higher than 0.90 was considered excellent. Test–retest reliability was calculated by determining the intraclass correlation coefficient (ICC) for the second PSQ-3-FI administration 7 ± 1 days later. ICC values ≤ 0.40 are considered to indicate fair reliability; 0.41–0.60, moderate reliability; 0.61–0.80, substantial reliability; and ≥0.81, excellent reliability. ICCs are reported with 95% confidence intervals (CI). Standard error of measurement (SEM) was calculated using the formula standard deviation × square root (1-ICC), where SD = the standard deviation for the change in PSQ-3 score from baseline to second administration. The smallest detectable change (SDC) was calculated using the formula SEM × 2. Confirmatory factor analysis (CFA) with ordinal variables in a one-factor model was used to investigate the validity of five variables with appropriate goodness-of-fit indices. Spearman’s correlation coefficients were used to investigate the convergent validity of the PSQ-3-FI by calculating the associations between total PSQ-3-FI scores and the scores on the CSI, TSK, DEPS, EQ-5D-5L, RMDQ, and pain history questions. The strengths of the correlations were interpreted as little or no correlation (Rs < 0.25), fair (0.25 > Rs ≤ 0.50), moderate to good (0.50 > Rs ≤ 0.75), or good to excellent (Rs > 0.75). The normality of the data was assessed using the Shapiro–Wilks and Kolmogorov–Smirnov tests. Group comparisons for normally distributed variables were performed using two-sample t-tests or repeated-measures ANOVA followed by the post hoc least significant difference (LSD) test. Categorical variables were compared using Fisher’s exact tests. The minimum required sample size for postural control comparison between groups was calculated with average means and estimated standard deviation from the review comparing pain-free controls and subjects with low back pain [47]. The two-tailed hypothesis was calculated on two independent study groups with 0.05 probability of type I error, 0.80 effect size, and 0.8 statistical power. The calculation revealed that at least 25 subjects had to be included in each group.

## 3. Results

### 3.1. Total Sample

There were no missing items in the data, as the electronic questionnaires automatically reminded the respondents if any items were missing.

Three subjects were excluded because they did not complete the study questionnaires as instructed during the clinical appointments. Five additional subjects were excluded as they had clear signs and symptoms of undiagnosed neurological pathological conditions affecting the central nervous system. Thus, the total number of participants included in this study was 249 (67 males and 162 females). Twenty of these participants only provided feedback on the face validity of the Finnish translation of the PSQ-3, and 229 subjects completed the psychometric validation portion of the study. The age range of the subjects was 20 to 65 years old (mean ± SD; 44.5 ± 11.7), and body mass index ranged from 25.6 to 45.2 (mean ± SD; 25.6 ± 4.8). The total cohort was divided into different subsamples for various analyses to test discriminative ability, postural control on the force plate, and sleep quality via actigraphy. The flow chart of subject recruitment and assessment is shown in Figure 1.

### 3.2. Reliability

Internal consistency (Cronbach’s alpha) was good (0.83 and ICC 0.61; 95% CI 0.55–0.68) and the ICC indicated test–retest reliability was excellent (0.91 and ICC 0.62; 95% CI 0.54–0.70). The inter-item correlation was 0.56 between items one and two, 0.73 between items two and three, and 0.53 between items one and three. The SEM for the change in PSQ-3 score between baseline and the second administration was calculated to be 1.28 and the SDC was 2.56. Details of the PSQ-3 scores and reliability data are presented in Table 1.

### 3.3. Structure Validation

The Kaiser–Meyer–Olkin measure (0.68) and Bartlett’s test of sphericity (*p* < 0.001) indicated that factor analysis was appropriate for this study sample. CFA analysis was carried out with ordinal variables to evaluate the fit of the indices to a one-factor model. As the PSQ-3 is a three-item PROM, the minimum number of items required for a one-factor model, no other factor model was considered. Additionally, the ratio of items per subject (1:76) exceeded the minimum ratio required (1:10) [52,53]. The recommended values for the fit indices and the CFA values for the one-factor model are presented in Table 2.

### 3.4. Convergent Validity of the PSQ-3-FI

As shown in Table 3, fair to good correlations were observed between the PSQ-3-FI and the Tampa Scale of Kinesiophobia, the Depression scale, and the 5-level EQ-5D. Moderate to good correlations were obtained between the PSQ-3-FI and the Roland–Morris disability questionnaire and Central Sensitization Inventory.

### 3.5. Discriminative Validity of the PSQ-3-FI

Of the total of 229 subjects, 42 (18.7%) reported no pain and were categorized as a control group. Specifically, the pain-free control subjects reported no CLBP, no radicular pain, a score of 0/10 on the pain scale, 0 months of pain history, no other chronic musculoskeletal pain, and no chronic headaches. The remainder of the subjects (187; 81.3%) reported chronic pain, of whom 161 (86%) reported CLBP. No subjects reported acute or subacute pain. Of the 187 subjects with chronic pain, 79 (34%) reported pain in a single body area only (CLBP group without leg referral or other chronic musculoskeletal pain or chronic headaches), and 108 (47%) reported multisite chronic pain (two or more of the following: CLBP with or without radiculopathy, other chronic musculoskeletal pain, and/or chronic headaches).

The comparisons of the demographics and clinical features of the groups presented in Table 4 revealed statistically significant differences in the PSQ-3-FI score between the pain-free control group, single chronic pain group, and multisite chronic pain group.

### 3.6. PSQ-3-FI Score Subgroups

The total cohort was divided into two subgroups according to PSQ-3-FI score, namely ≤4 (48%; *n* = 110) or ≥5 (52%; *n* = 119). The only significant demographic difference between these groups was age, with the ≥5 subgroup being older (mean ± SD; 42.6 ± 10.9 vs. mean ± SD; 46.2 ± 12.2; *p* = 0.03). None of the subjects reported dizziness at the same time the postural control tests were carried out.

In the PSQ-3-FI score ≤ 4 group, 18 subjects reported dizziness and 92 reported no dizziness (dizziness:no dizziness ratio: 1:5.1) over the previous 12 months. In the PSQ-3-FI score ≥ 5 group, 34 subjects reported dizziness and 84 reported no dizziness (ratio: 1:2.4) over the previous 12 months. Group comparisons showed the prevalence of dizziness was significantly higher (*p* = 0.03) in the higher PSQ-3-FI score group. The subtypes of dizziness were classified into seven classifications (Table 5).

The results of postural control tests on the force plate for the PSQ-3-FI score ≤ 4 or ≥ 5 groups are compared in Table 6. The higher-score PSQ-3-FI group consistently exhibited (in 12 out of 12 tests) greater postural control impairment, with significant differences observed between the two PSQ-3 groups in the majority (7/12; 58%) of tests. Significant differences were observed between groups for all tests performed with eyes open (6/6; 100%). The only significant difference between groups in eyes-closed tests was observed on the stable surface (Figure 2).

### 3.7. Objective Sleep Quality

The parameters representing sleep quality are presented in Table 7 for the lower and higher PSQ-3-FI score groups. The average awakening length was longer in the group with higher PSQ-3-FI score (PSQ-3-FI ≥ 5) compared to the group with lower PSQ-3-FI score. The other four sleep quality parameters, total sleep time, sleep efficiency, and number of awakenings lasting more than one of five minutes, were not significantly different between the groups with higher and lower PSQ-3 scores.

## 4. Discussion

The reciprocal relationship between pain affecting sleep—and disturbances to sleep affecting pain—is well-established [2,3,4,5,9,10,11]. However, the PSQ-3 is the only PROM for directly assessing pain–sleep interactions in the CLBP population [13]. It appears that the study of the direct sleep–pain interaction is neglected in almost all clinical musculoskeletal chronic pain studies, and the short, reliable, valid PSQ-3 has not been widely used in clinical assessment or research. This might be because there are quite a large number of similar, but improperly validated pain–sleep PROMs [13]. Additionally, clinical studies on CLBP over the last decades have paid little attention to the assessment and outcome measurements for sleep [54,55].

The results of this validation reveal the acceptable measurement properties of the PSQ-3 for the CLBP population; therefore, we propose a more widespread implementation of the PSQ-3 for clinical assessments, and research will enable the measurement of the direct effect of pain on sleep. Moreover, a meta-analysis published in 2018 lends further clinical validity to the items of the PSQ-3, as gold-standard objective polysomnographic measures of sleep among a population with chronic musculoskeletal pain syndromes concluded these individuals “experience significant sleep disturbances, particularly concerning sleep initiation and maintenance” [3], which are the exact features measured by the PSQ-3 [12].

### 4.1. Reliability

Internal consistency (Cronbach’s alpha) indicates how well PROM items correlate with and predict each other. The total inter-item correlation and prediction for the PSQ-3-FI were rated as good (0.83) and single inter-item correlations were moderate to good. The highest correlation was observed between items “2. How often have you been awakened by pain during the night?” and “3. How often have you been awakened by pain in the morning?”. Moreover, test–retest reliability was excellent (0.91). Interestingly, the PSQ-3 questions consider symptoms over the previous week, which is identical to the test–retest evaluation timeframe (7 ± 1 days). Hence, we conclude that symptom recall over one week appears to be a reliable timeframe for the majority of the subjects in this study. The inter-item correlation between items one and two was 0.56, 0.73 between items two and three, and 0.53 between items one and three; these values compare well with the English version, for which the inter-item correlations were 0.65, 0.73, and 0.58, respectively [12]. The SEM value indicates the likelihood of a “true” score that represents a reliable score without any fluctuations due to systematic and random factors related to the measurement process. The general rule is that lower SEM values indicate higher reliability and more confidence that the score has been measured accurately. The SDC is defined as the change in the instrument’s score beyond measurement error [56]. The SEM value of 1.28 and SDC value of 2.56 indicate that the results were measured fairly accurately, without fluctuations due to systemic or random factors related to the measurement process. Based on the SDC, we can be confident that the observed change is real, as a minimum change of 2.56 on a scale from 0 to 10 needs to be observed. Internal consistency was 0.87 in the previous PSQ-3 validation [12], in line with our value of 0.83. Test–retest reliability, SEM, and SDC were not calculated in the previous validation [12]; hence, comparison of these values is not possible.

### 4.2. Structure Validation

Model fit indices with CFA can be used for test to accept or refute the proposed factor model [57]. Chi-Square, CFI, TLI, RMSEA, and SRMR are the most commonly reported fit indices of CFA [58]. In our results, model fit indices showed acceptable fit indices for a one-factor model, except for the Chi-Square test. However, it is well known that the larger the sample size, the greater the chances of obtaining a statistically significant Chi-square result [59]. Hence, rejection of the Chi-Square is probably due to the Chi-Square test being insensitive to larger sample sizes as in this study. Overall, fit indices indicated an acceptable fit for the one-factor model. CFI and TLI indicated a perfect fit compared to the PSQ-3 English version, though the Chi-Square and RMSEA tests showed non-acceptable fits. Overall, one-factor model also represented the best fit for the original version, where the PSQ-3 was shortened to only include three items [12].

### 4.3. Convergent Validity

The relationship of PSQ-3-FI total score with PROMs showed a fair correlation of TSK, EQ-5D-5L, and DEPS and moderate to good correlation with RMDQ and CSI. The DEPS and CSI include questions that directly assess the quality of sleep [32,38]. In the previous validation of the PSQ-3, the Pain and Disability Index, Pain Intensity Questionnaire, and SF-36 Short Form Health Survey pain-related questionnaires exhibited very similar correlations with the total PSQ-3 score [12]. The total PSQ-3-FI scores showed a fair correlation with pain duration in months and dichotomous yes/no answers on chronic low back pain, pain referring to leg, other chronic musculoskeletal pain, and pain duration in months. Chronic headache and dizziness in the past 12 months showed little or no correlation with PSQ-3-FI total score. The numerical pain scale was the only dichotomous variable that showed moderate to good correlation with the total PSQ-3 score. The tight correlation between sleep quality and next-day pain intensity is well established, with previous studies showing a clear relationship between poorer sleep and short-term increases in pain intensity in patients with chronic pain [60,61]. The correlations with pain and disability PROMs were generally at the same level in this study and the English version; however, quality of life was better in this study. It must be noted that such comparisons are not reliable due to the variability across the measurement properties of different PROMs.

### 4.4. Discriminative Validity

There are no previous studies of the discriminative validity of any PROMs that assess the effect of pain on sleep [13]. However, the total PSQ-3-FI score could distinguish pain-free controls, subjects with chronic pain in a single body area, and subjects with multisite chronic pain. The factors leading to chronic pain are complex, with multiple contributors leading to persistent symptoms [1]. Sleep quality and disturbances are one of the main contributors to chronic musculoskeletal pain [10,11]. The significant discriminative ability of the total PSQ-3-FI score to identify controls with no pain and the -ite and multisite chronic pain groups is in line with previous research that showed clear relationships between multisite pain and various sleep disturbances [62,63].

### 4.5. Relationships of PSQ-3-FI Score with Subjective Dizziness, Postural Control, and Sleep Quality

We observed a higher prevalence of dizziness in the higher PSQ-3-FI score group. A single study that explored the relationship between sleep disturbances and dizziness reported an increase in the prevalence of dizziness symptoms among subjects with a variety of sleep disturbances (16).

We also found little total sample correlation but consistent relationships between the PSQ-3-FI score and a range of postural control parameters, indicating poorer postural control among the group with higher PSQ-3-FI score. Overall, greater postural instability was observed during the more challenging eyes-closed rather than eyes-open conditions, and more so on an unstable surface. These statistically significant intergroup differences (PSQ-3-FI ≤ 4 vs. PSQ-3-FI ≥ 5) were observed in the majority of tests. Perhaps surprisingly, significant differences between groups were observed for all eyes open tests, but in only one of the six eyes-closed tests. This is especially surprising, as the majority of subjects in our study were suffering from CLBP. Previous studies of the CLBP population have shown a clear relationship between postural control impairments that are most marked when the subject’s eyes are closed [64,65,66]. This clinical phenomenon is explained by the sensory weighting theory of postural control, which suggests that somatosensory, vestibular, and visual sensory information have mutual effects on postural control and are weighted as a sum of parts [67]. Subjects with CLBP exhibit impaired somatosensory information processing (the ability to feel through muscle, joint, and fascia-based sensory receptors) and hence this sensory channel is “down-weighted” due to pain and associated physical disabilities. Moreover, this altered sensory weighting causes increased postural sway in eyes-closed tests due to an inability to compensate for impaired somatosensory information using visual cues. Naturally, postural control is not simply just the sum of the weighting of sensory information. Sensorimotor integration in the central nervous system acts in conjunction with these raw sensory signals to ultimately govern postural control in a context-dependent manner [68,69]. Sleep disturbances, such as sleep deprivation, are well-known to negatively influence central nervous system processes across multiple cognitive [8], emotional [6], and motor functions [7], and in turn, vestibular pathologies can induce sleep disturbances [17]. The reciprocal effect of sleep disturbances upon such motor and cognitive processes may be one explanation for this seemingly paradoxical finding. In contrast, there was just one significant difference between groups out of the six parameters assessed under eyes-closed conditions. The fact that the higher PSQ-3 score group reported significantly more dizziness symptoms argues in favor of such a reciprocal relationship between vestibular dysfunction and sleep disturbances. Unfortunately, the mixed results obtained after subgrouping dizziness into seven subtypes did not help to further elaborate our findings.

Regarding sleep quality, the average awakening length was longer in the higher total PSQ-3 score group than the lower total PSQ-3 score group. Only minor differences were observed for the other sleep quality measurements, though there was a general trend towards poorer sleep quality among the higher PSQ-3 score group. These differences between groups (PSQ-3-FI ≤ 4 vs. PSQ-3-FI ≥ 5) may be affected to at least some degree by the long administration period between the PSQ-3 and actigraphy, which could have been up to 18 months. This was partly due to the COVID-19 outbreak in Finland, when data collection was stopped for five months. PSQ-3 test–retest reliability was excellent over one week; however, reliability may obviously vary after more than one year. Gender differences may also evidently affect the group comparisons, as females generally exhibit better sleep parameters in actigraphy [70]. The lower PSQ-3 score group contained 40% males and the higher PSQ-3 score group did not include any males. Moreover, the group with higher PSQ-3 score was, on average, more than ten years older than the lower-score group, which can be theorized to have a counter-intergroup effect on the results, as sleep quality parameters generally correlate negatively with aging [71]. Despite some limitations in relation to the long administration time and gender differences between groups, actigraphy may represent a relevant component of future pain–sleep PROM validations. Indeed, previous studies reported similar general trends between higher PROM scores and poorer sleep quality with other pain–sleep-related questionnaires (25, 26). We therefore conclude that there is an evident trend towards poorer sleep quality among individuals with higher PSQ-3-FI scores.

### 4.6. Strengths and Limitations

#### 4.6.1. Strengths

Some strengths of this study include the very thorough validation of different measurement properties of the PSQ-3, including cross-cultural validity, face validity, internal consistency, test–retest reliability, measurement error, discriminant validity, and convergent validity, as well as the adequate size of the subject cohort and control group. Furthermore, the relationships between three novel measurements and PSQ-3 scores were investigated: subjective symptoms of dizziness, postural control testing, and sleep quality by actigraphy. Moreover, this is the first study to assess the exact effect of pain on sleep in relation to postural control between two groups of subjects. As far as we are aware, this study is the most comprehensive validation of any of the existing twelve PROMs that assess the effect of pain on sleep [13].

#### 4.6.2. Limitations

As with other studies of this kind, our results are based on a single cohort assessed at a single clinic, so generalization to other subject populations should be made with caution. All symptoms were self-reported, and pain reporting was limited by the items on a single questionnaire. Actual medical diagnoses by a trained clinician were lacking. However, the self-reported data from our subject sample showed no discrepancies or illogical patterns of answers to suggest the results were invalid or had any considerable negative effects on our findings. Furthermore, it should also be noted that diagnoses of different types of musculoskeletal pain by trained clinicians are mostly based on patient self-reporting [72]. The most notable data collection limitation was the rather long time period between PSQ-3 administration and actigraphy.

#### 4.6.3. Suggestions for Further Research

Future studies could validate the PSQ-3 in different musculoskeletal pain populations, such as individuals with neck pain and central sensitization syndromes such as fibromyalgia. Due to the high acceptable objective reliability of actigraphy for sleep parameters, actigraphy could be more extensively implemented in future validations of the PSQ-3 among larger cohorts of subjects and over longer time periods. Naturally, the only path towards more universal use of PSQ-3 is following cross-cultural validations.

## 5. Conclusions

The Finnish translation of the PSQ-3 was successfully cross-culturally adapted and validated. The measurement properties of the PSQ-3-FI were all acceptable for the Finnish-speaking CLBP population. Additional studies that implement the PSQ-3 as a short, valid, reliable instrument for screening assessments and outcome measurements could lead to the development of a better understanding of the direct effect of multifactorial musculoskeletal pain on sleep.

## Figures and Tables

**Figure 1 jcm-10-04887-f001:**
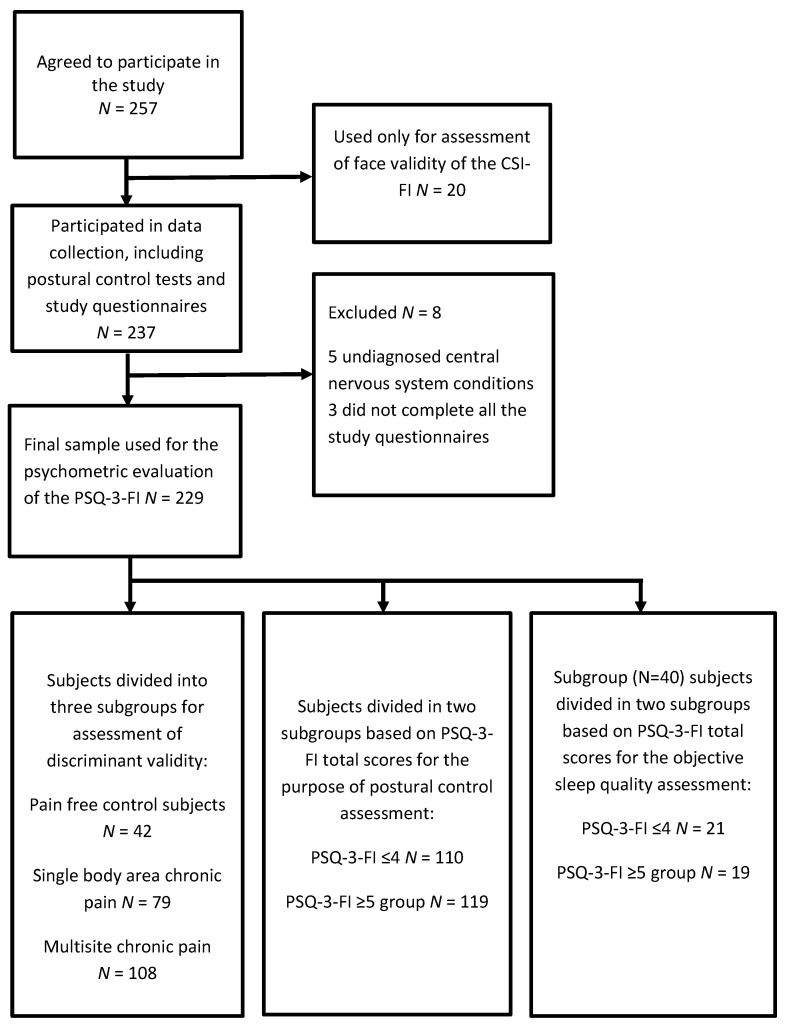
Flow chart of subjects.

**Figure 2 jcm-10-04887-f002:**
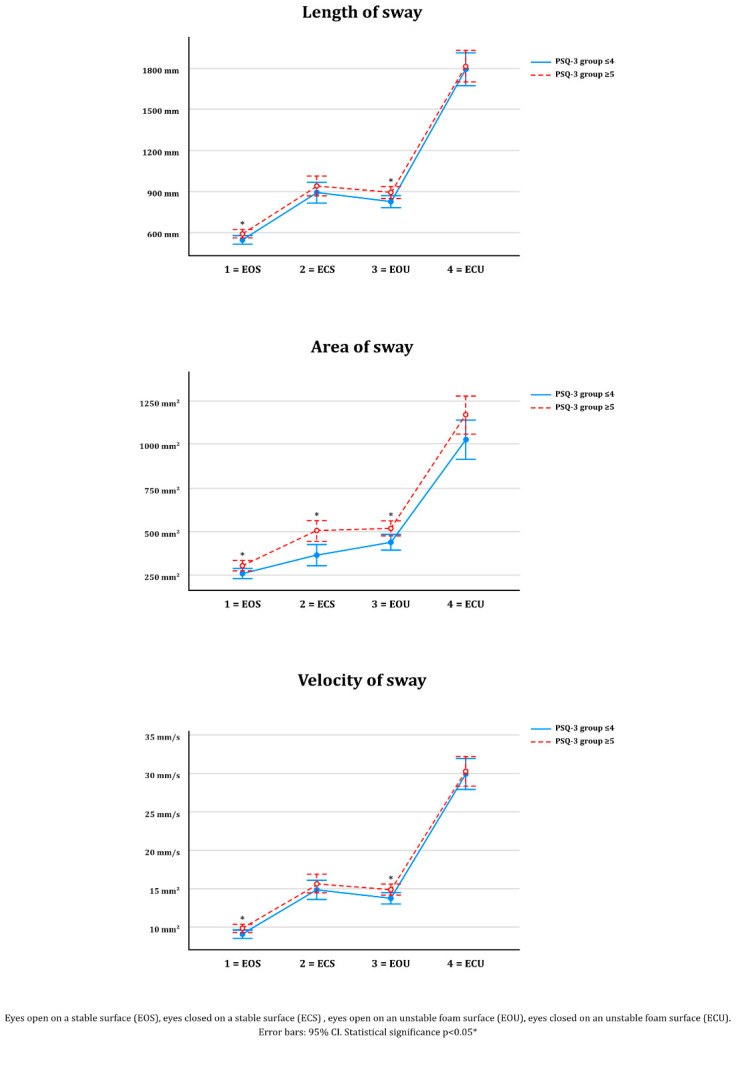
Postural control on PSQ-3-FI groups score ≤4 or ≥5 groups (*n* = 229).

**Table 1 jcm-10-04887-t001:** Mean Pain and Sleep Questionnaire three-item index (PSQ-3-FI) scores.

Question	Item	Mean (95% CI)	Range	ICC
1	How often have you had trouble falling asleep because of pain?	2.5 (2.1–2.8)	0–9	0.55
2	How often have you been awakened by pain during the night?	2.4 (2.0–2.8)	0–10	0.52
3	How often have you been awakened by pain in the morning?	2.0 (1.7–2.4)	0–10	0.74
Total		6.9 (6.0–7.8)	0–32	0.61

Confidence interval (CI); Intraclass correlation (ICC).

**Table 2 jcm-10-04887-t002:** Confirmatory factor analysis fit indices for the one-factor model.

Fit Index	Recommended Value	Value	Not Acceptable/Acceptable
Chi-Square	>0.05	*p*-value, 0.041	Not acceptable
CFI	>0.95	0.985	Acceptable
TLI	>0.95	0.978	Acceptable
RMSEA	<0.05	ICC 0.098; 95% CI 0.017–0.188; *p*-value 0.127	Acceptable
SRMR	<0.08	0.07	Acceptable

Comparative fit index (CFI); Tucker–Lewis index (TLI); Root mean square error of approximation (RMSEA); Standardized Root Mean Square Residual (SRMR).

**Table 3 jcm-10-04887-t003:** Correlations between total PSQ-3-FI scores and subject-reported outcome measures, history, and postural control test parameters (*n* = 229).

Clinical Variables	Correlation (ρ) with PSQ-3 Total Score
**Subject-reported outcome measures**	
Tampa scale of kinesiophobia (TSK)	0.41 *
Depression scale (DEPS)	0.305 *
The Roland–Morris disability questionnaire (RMDQ)	0.54 **
The Central Sensitization Inventory (CSI)	0.51 **
EuroQol The 5-level EQ-5D version (EQ-5D-5L)	−0.44 *
**History**	
Chronic low back pain	0.39 *
Pain referral to leg	0.32 *
Other chronic musculoskeletal pain	0.31 *
Numerical pain scale	0.50 *
Pain duration in months	0.33 *
Chronic headache	0.18
Dizziness in past 12 months	0.16
**Postural control EOS**	
Length of sway	0.13
Area of sway	0.13
Velocity of sway	0.13
**Postural control ECS**	
Length of sway	0.08
Area of sway	0.20
Velocity of sway	0.08
**Postural control EOU**	
Length of sway	0.12
Area of sway	0.19
Velocity of sway	0.12
**Postural control ECU**	
Length of sway	0.02
Area of sway	0.14
Velocity of sway	0.02

The Spearman’s rank correlation coefficient (ρ). Little or no correlation (ρ < 0.25), fair correlation (0.25 > ρ ≤ 0.50) *, moderate to good correlation (0.50 > ρ ≤ 0.75) **. Eyes open on a stable surface (EOS); eyes closed on a stable surface (ECS); eyes open on an unstable foam surface (EOU); eyes closed on an unstable foam surface (ECU).

**Table 4 jcm-10-04887-t004:** Demographics and discriminative validity of the PSQ-3 among the three subgroups (*n* = 229).

	Pain-Free Control Group (*n* = 42)	Chronic Pain in a Single Body Area (*n* = 79)	Multisite Pain (Two or More Chronic Pain Locations) (*n* = 108)	*p*-Value One-Way ANOVA	Comparison between Three Groups; Post Hoc LSD
Age (years)	40.2 ± 10.6	44.5 ± 11.6	46.1 ± 12.0	0.02 *	b (0.005 *)
Male/female (n/n)	13/29	31/48	23/85	0.03 *	c (0.008 *)
Height (cm)	171.3 ± 8.2	172.6 ± 9.9	170.1 ± 9.2	0.18	
Weight (kg)	76.6 ± 15.9	76.1 ± 19.3	74.4 ± 15.4	0.71	
BMI	26.0 ± 4.8	25.2 ± 4.8	25.8 ± 4.9	0.66	
PSQ-3-FI	2.0 ± 3.6	6.2 ± 6.6	9.3 ± 7.4	<0.001 *	a (0.01 *) b (<0.001 *) c (0.01 *)

One-Way ANOVA post hoc comparison based on Fisher’s Least Significant Difference (LSD). Statistical significance, *p* < 0.05 *. Comparison between control group without pain and single body area pain (a), between the control group without pain and multisite pain (b), and between single body area pain and multisite pain (c). Body mass index (BMI).

**Table 5 jcm-10-04887-t005:** Comparison of the PSQ-3-FI score ≤ 4 and ≥ 5 groups (*n* = 229).

	PSQ-3-FI ≤ 4 (*n* = 110)	PSQ-3-FI ≥ 5 (*n* = 119)	*p*-Value
Age (years)	42.6 ± 11.0	46.2 ± 12.2	0.02 *
Height (cm)	171.1 ± 8.8	171.3 ± 9.8	0.85
Weight (kg)	74.3 ± 15.3	76.4 ± 18.3	0.34
BMI	25.3 ± 4.7	25.9 ± 5.0	0.33
PSQ-3-FI	1.22 ± 1.4	12.1 ± 6.1	0.001 *
Male/female (n/n)	35/75	32/87	0.47
Dizziness in past 12 months, no/yes	92/18	85/34	0.03 *
Dizziness subtypes, *n* (%)			
1. Off balance or unsteadiness	4 (22%)	2 (6%)	
2. Light-headedness	7 (40%)	5 (15%)	
3. Feeling as if passing out	0	8 (24%)	
4. Spinning or vertigo	2 (10%)	4 (12%)	
5. Floating or tilting sensation	5 (28%)	12 (34%)	
6. Blurring of vision when moving the head	0	3 (9%)	
7. Other	0	0	

One-Way ANOVA and Fisher’s exact tests; statistical significance, *p* < 0.05 *. Body mass index (BMI).

**Table 6 jcm-10-04887-t006:** Postural control on the force plate for the PSQ-3-FI score ≤ 4 and ≥ 5 groups (*n =* 229).

Test	PSQ-3 Group	Mean ± SD	*p*-Value
Length of sway (mm)			
EOS	≤4	547 ± 150	0.048 *
	≥5	592 ± 186	
ECS	≤4	891 ± 452	0.34
	≥5	940 ± 349	
EOU	≤4	826 ± 220	0.04 *
	≥5	860 ± 240	
ECU	≤4	1794 ± 645	0.80
	≥5	1805 ± 638	
Area of sway (mm^2^)			
EOS	≤4	258 ± 138	0.04 *
	≥5	303 ± 178	
ECS	≤4	364 ± 253	0.002 *
	≥5	503 ± 383	
EOU	≤4	438 ± 218	0.012 *
	≥5	518 ± 260	
ECU	≤4	1029 ± 540	0.08
	≥5	1170 ± 659	
Velocity of sway (mm/s)			
EOS	≤4	9.1 ± 2.5	0.048 *
	≥5	9.9 ± 3.1	
ECS	≤4	14.9 ± 7.5	0.35
	≥5	15.7 ± 5.8	
EOU	≤4	13.8 ± 3.7	0.036 *
	≥5	14.9 ± 4.2	
ECU	≤4	29.9 ± 10.7	0.802
	≥5	30.3 ± 10.6	

Eyes open on a stable surface (EOS); eyes closed on a stable surface (ECS); eyes open on an unstable foam surface (EOU); eyes closed on an unstable foam surface (ECU). Repeated measures ANOVA and post hoc Least Significant Difference (LSD); statistical significance, *p* < 0.05 *.

**Table 7 jcm-10-04887-t007:** Sleep quality measures based on actigraphy for the PSQ-3-FI score ≤ 4 and ≥ 5 groups (*n* = 40).

	PSQ-3-FI ≤ 4 (*n* = 21)	PSQ-3-FI ≥ 5 (*n* = 19)	*p*-Value
Age (years)	41.6 ± 12.2	51.9 ± 10.8	0.008 *
Height (cm)	170.5 ± 8.23	169.7 ± 7.6	0.77
Weight (kg)	74.6 ± 14.1	74.2 ± 12.5	0.93
BMI	25.6 ± 3.8	25.8 ± 4.2	0.87
PSQ-3-FI	1.6 ± 1.9	12.6 ± 7.1	0.001 *
Male/female (n/n)	6/15	0/19	0.02 *
Total sleep time (min)	401.8 ± 58.7	393.4 ± 77.7	0.70
Sleep efficiency (%)	86.1 ± 4.1	83.5 ± 7.4	0.18
Number of Awakenings > 1 min	22.8 ± 8.0	22.6 ± 10.6	0.94
Average awakening length (min)	2.6 ± 0.7	3.3 ± 1.2	0.03 *
Number of awakenings > 5 min	4.3 ± 3.0	5.8 ± 3.6	0.16

Statistical significance, *p* < 0.05 *.

## Data Availability

The data presented in this study are available on request from the corresponding author. The data are not publicly available due to containing information that could compromise the privacy of research participants.

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
