# Peer review of "Cross-Cultural Adaptation, Reliability, and Psychophysical Validation of the Pain and Sleep Questionnaire Three-Item Index in Finnish"

_jcm, 2021, doi:10.3390/jcm10214887_

Round 1
Reviewer 1 Report
Thank you for sharing this manuscript with me.
The authors put a lot of effort into it, and it is very well-designed' and follows the publication rules.
The aim of this study was to translate and cross-culturally adapt the PSQ-3 into Finnish and to evaluate its properties.
Since the tool of "PSQ-3 had good internal consistency and good structural validity for a one-factor model, but doubtful convergence validity" (line 62), the new PSQ-3-FI tries to improve its validity. "Evaluate its reliability and structural validity" (line 95).
The cross-cultural validation is detailed and written in a clear language, and the results show good validity.
Some minor remarks:
Abstract
The aim of the study is not clearly mentioned.
Please add.
Material and Methods:
Line 160- Actigraphy data were collected between December 2019 to March 2020 and August 2020 to November 2020. That is a limitation of the study as written due to the Covid-19 outbreak.
When were the data of the PSQ-3 collected? Was there a difference between the group without a time gap and the group with?
Results:
Line 235-
Why did the authors choose this cut-off? Please add.
Author Response
Please see the attachment.

Reviewer 2 ReportThis manuscript has high quality and relevant interest for the readers.
I have a question related to the sample size. What is the sample size calculation? and the justification of the sample recruited? Do you consider the representation of the population? You have included different pathologies and you should be cautious about the sample selection.
My second question is according to the second objective, the relationship that you explain in the introduction. You compared the groups but you did not analyze a correlation, please explain this aspect and change it in the manuscript.
